# Development of a mugineic acid family phytosiderophore analog as an iron fertilizer

Motofumi Suzuki [1], Atsumi Urabe[2], Sayaka Sasaki[2], Ryo Tsugawa[2], Satoshi Nishio[2], Haruka Mukaiyama[2], Yoshiko Murata[3], Hiroshi Masuda [4,8], May Sann Aung [4,8], Akane Mera[1], Masaki Takeuchi [2], Keijo Fukushima[2], Michika Kanaki[5], Kaori Kobayashi [5], Yuichi Chiba [6], Binod Babu Shrestha[2], Hiromi Nakanishi [6], Takehiro Watanabe[3], Atsushi Nakayama[2], Hiromichi Fujino[2], Takanori Kobayashi[4], Keiji Tanino [7], Naoko K. Nishizawa[4,6] & Kosuke Namba [2✉]

Iron (Fe) is an essential nutrient, but is poorly bioavailable because of its low solubility in alkaline soils; this leads to reduced agricultural productivity. To overcome this problem, we first showed that the soil application of synthetic 2′-deoxymugineic acid, a natural phytosiderophore from the Poaceae, can recover Fe deficiency in rice grown in calcareous soil. However, the high cost and poor stability of synthetic 2′-deoxymugineic acid preclude its agricultural use. In this work, we develop a more stable and less expensive analog, proline-2′-deoxymugineic acid, and demonstrate its practical synthesis and transport of its Fe-chelated form across the plasma membrane by Fe(III)•2′-deoxymugineic acid transporters. Possibility of its use as an iron fertilizer on alkaline soils is supported by promotion of rice growth in a calcareous soil by soil application of metal free proline-2′-deoxymugineic acid.

[1] Aichi Steel Corporation, Tokai-shi, Aichi, Japan. [2] Department of Pharmaceutical Sciences, Tokushima University, Tokushima, Japan. [3] Bioorganic Research Institute, Suntory Foundation for Life Sciences, Soraku-gun, Kyoto, Japan. [4] Ishikawa Prefectural University, Nonoichi, Ishikawa, Japan. [5] Graduate School of Chemical Sciences and Engineering, Hokkaido University, Sapporo, Japan. [6] The University of Tokyo, Bunkyo-ku, Tokyo, Japan. [7] Faculty of Science, Hokkaido University, Sapporo, Japan. [8] Present address: Akita Prefectural University, Akita, Japan. ✉email: namba@tokushima-u.ac.jp

Globally, over one-third of all land is covered by high-pH alkaline or calcareous soils, which are unfavorable for agricultural use[1,2]. In such soils, most plants develop iron (Fe)-deficiency chlorosis, which prevents normal growth. Despite the high total Fe content of alkaline soils, most plants have difficulty absorbing the Fe because it exists as water-insoluble ferric-hydroxide or ferric-oxide. To overcome this problem, members of the Poaceae have developed a sophisticated mechanism of Fe acquisition through secretion of mugineic acid family phytosiderophores (MAs) into the rhizosphere. These natural chelators form 1:1 water-soluble complexes with Fe(III), and Poaceae plants can absorb the resulting Fe(III)–MAs complexes via yellow stripe 1/yellow stripe 1-like (YS1/YSL) transporters (Fig. 1) (i.e., strategy-II)[3–8]. Secretion of MAs increases in response to Fe deficiency stress and is positively correlated with tolerance of low Fe availability, but differs among the Poaceae[9]. Rice is particularly sensitive to Fe deficiency stress because of its poor secretion of 2′-deoxymugineic acid (DMA) (**1**), which is a natural Fe chelator in rice[10]. Therefore, transgenic rice lines harboring genes from barley involved in the synthesis of MAs have been developed, which exhibit improved tolerance to low Fe availability in alkaline soils[11,12]. However, this transgenic approach is limited to species amenable to transformation techniques, and requires stabilization of traits in lines adapted to the many specific environments in which cereals are grown.

External application of Fe(III)–MAs may be an effective and practical alternative method to overcome Fe deficiency. Indeed, soil application of synthetic Fe chelates is the most commonly applied technique in agriculture, and the development and application of Fe chelates in alkaline soil have been intensively studied[13]. Non-poaceae plants effectively utilize artificial aminopolycarboxylate chelating agents such as Fe-ethylenediaminetetraacetic acid (Fe-EDTA), Fe-diethylenetriaminepentaacetic acid (Fe-DTPA), and the especially stable Fe complex of o,o-ethylendiamine di(o-hydroxyphenylacetic) acid (Fe-EDDHA) for Fe uptake. Thus, Fe(III)-chelate complexes are enzymatically reduced at the root surface and the Fe(II) released from the chelates is taken up by transporters (i.e., strategy-I). However, the considerable environmental threat conferred by persistence of these chelating agents remains a concern[14]. Moreover, in Poaceae plants (i.e., strategy-II) that absorb the Fe(III)-MAs complexes, these artificial chelating agents are ineffective fertilizers with respect to Fe uptake[15]. Thus, there is a need for the development of biodegradable Fe chelates that are effective for all species[16,17]. Natural Fe chelator MAs are expected to be more effective than conventional artificial chelates for Poaceae in alkaline soil. Furthermore, because it has been suggested that non-Poaceae plants can also utilize MAs secreted from Poaceae in intercropping fields[18], MAs may be useful chelators for all plant species, including fruit crops[19]. However, published studies concerning the effects of purified MAs have used only hydroponic cultures[20–22]. To our knowledge, there

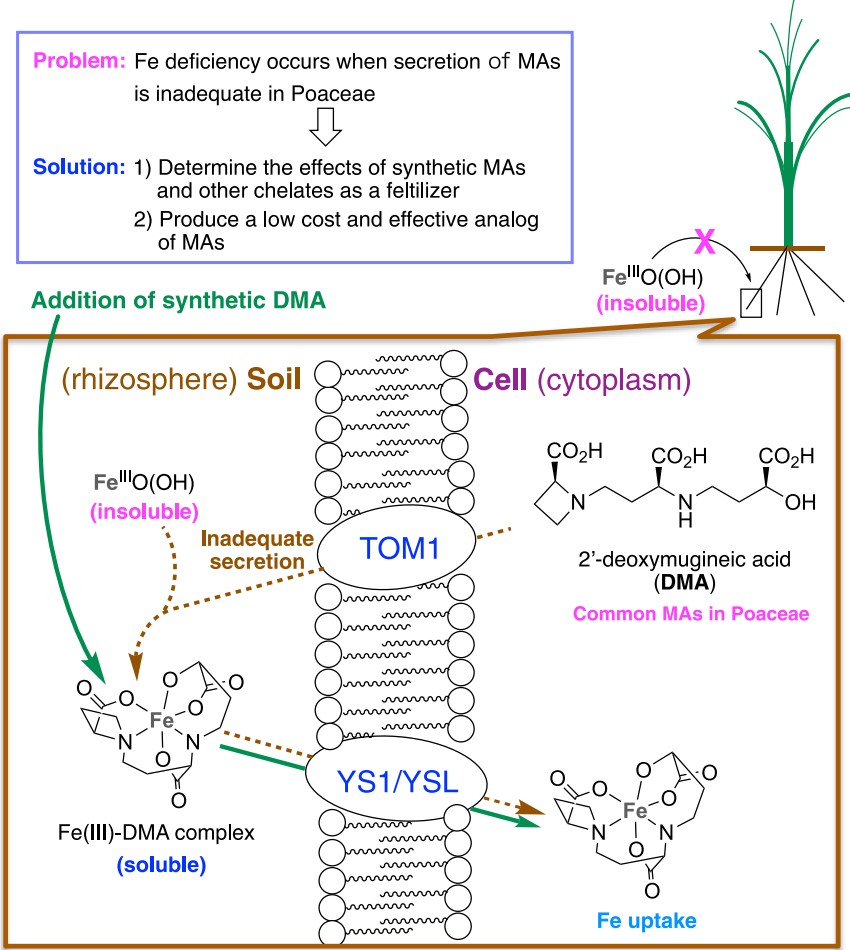

**Fig. 1 Iron (Fe) uptake strategy in Poaceae plants.** Poaceae plants secrete mugineic acid family phytosiderophores (MAs) into the rhizosphere and absorb Fe-MAs complex. Fe deficiency occurs when secretion of MAs is inadequate. This problem may be overcome by adding synthetic 2′-deoxymugineic acid (DMA) or an analog thereof. DMA 2′-deoxymugineic acid, MAs mugineic acids, TOM1 transporter of mugineic acid 1, YS1/YSL yellow stripe 1/yellow stripe 1 like transporter.

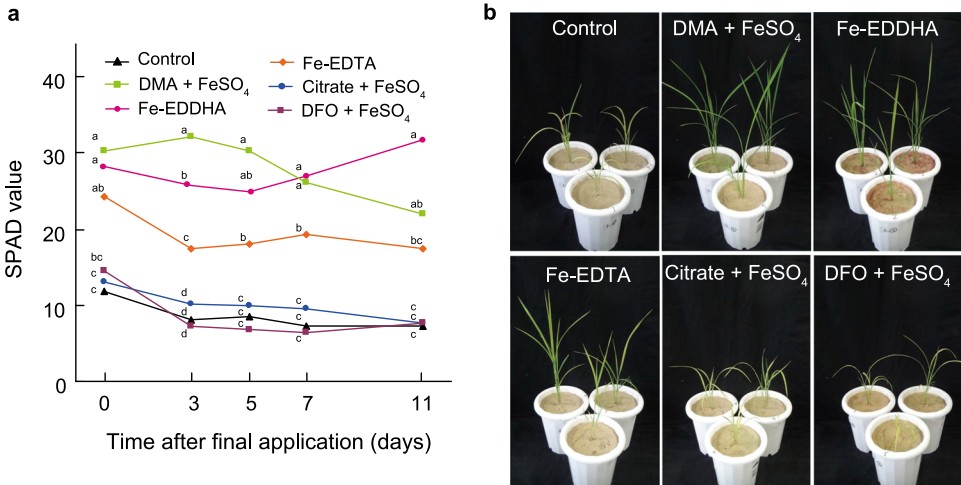

**Fig. 2 Effects of DMA application on plant growth in calcareous soil. a** Soil and plant analyzer development (SPAD) values of the newest rice leaves after the last of six applications of Fe-chelates at 14 days after transplantation (0 days after the last application). The chelates (DMA, citrate, and desferrioxamine [DFO]) were mixed with equimolar $FeSO_4$ in water before application. Values are means of three replicates in which each replicate is average of three plants in each pot, and different letters at the same time point indicate significant differences at $P < 0.05$ by Tukey's honestly significant difference test (two-sided). $P$-values for one-way analysis of variance (ANOVA) are follows: 0 days, 0.0001; 3 days, <0.0001; 5 days, <0.0001; 7 days, <0.0001; 11 days, <0.0001. **b** Rice plants at 3 days after the last application. DMA 2'-deoxymugineic acid, EDTA ethylenediaminetetraacetic acid, EDDHA ethylenediamine-N,N'-bis(2-hydroxyphenylacetic acid), DFO desferrioxamine, SPAD soil and plant analyzer development.

has been no report of the successful use of MAs in soil application, probably due to their high supply cost and instability in soils[23]. Thus, the possibility of using MAs for agricultural applications has been overlooked in the 40 years since their discovery[3].

Here, by fusion of synthetic organic chemistry and plant physiology, we report the effect of synthetic DMA, a common phytosiderophore in Poaceae (e.g., rice, corn, and wheat), in a growth experiment using rice as a model plant in calcareous soil. We also describe the development and application of an innovative derivative of DMA as a fertilizer with improved stability and reduced cost.

## Results

**Effects of DMA application to calcareous soil**. The rice plant is sensitive to Fe deficiency because of its low DMA secretion and remains difficult to grow on alkaline soils, although the whole rice genome has been decoded and considerable rice-related molecular biology information has been accumulated. Thus, the rice plant is the most suitable model plant to evaluate the efficacy of MAs, based on research in transgenic plants[11,12]. We first examined the effect of addition of 30 μM Fe-DMA as a solution to calcareous soil compared with other Fe-chelating agents including citric acid and desferrioxamine (DFO). Synthetic Fe chelating agents, namely Fe-ethylenediaminetetraacetic acid (EDTA) and commercial Fe-ethylenediamine-N,N'-bis(2-hydroxyphenylacetic acid) (EDDHA), as well as water control, were also compared. DMA for use in growth experiments was generated through an efficient synthesis method previously developed in our laboratory[24]. The calcareous soil was mined from a site analyzed previously[25] and virgin soil was used in all experiments. Citric acid is secreted by the roots of higher plants, DFO is a siderophore secreted from the bacterium *Streptomyces pilosus* for Fe acquisition[26], EDTA is a commonly used synthetic chelator of minerals, and commercial Fe-EDDHA contains the highly stable Fe-*o*, *o*-EDDHA complex in combination with regioisomers of low-stability phenolic hydroxy complexes. Seven-day-old seedlings were transplanted to calcareous soil, and mixtures of Fe and chelate were added six times at from 3 to 14 days (approximately every 2 days) after transplantation to

mitigate the effects of biodegradation. The six applications of chelates were completed on the 14th day, and then evaluations of the Fe status of the rice plants were initiated (0 days in Fig. 2a). The Fe status was measured by the Soil and Plant Analyzer Development (SAPD) values of the newest leaves, which are highly correlated with chlorophyll content[27,28]. Usually, Fe-sufficient green rice leaves have SPAD values of around 40, and severe Fe chlorotic leaves have SPAD values below 20.

At 14 days after transplantation (the day of the final Fe-chelate application; 0 days in Fig. 2a), the rice plants without Fe addition (control) showed severe Fe deficiency with SPAD values around 12. In contrast, the SPAD values were around 30 in plants with Fe-DMA, which was significantly higher than the control value (Fig. 2a). Both Fe-EDDHA and Fe-EDTA moderately improved the chlorosis, but Fe-citrate and Fe-DFO were ineffective. With no further application of Fe-chelates to the soil after 14 days, the SPAD values of the leaves of plants supplied with Fe-DMA were significantly higher over the next 5 days, compared with the values of plants supplied with Fe-EDDHA or Fe-EDTA (Fig. 2a, b). However, the SPAD values of plants fertilized with Fe-DMA decreased thereafter, but those of plants fertilized with Fe-EDDHA were enhanced (Fig. 2a). These differences were likely due to variations in the degradability of these chelators in soil. In particular, MAs are readily degraded by soil microorganisms[23], but Fe-EDDHA is stable in calcareous soils[29]. Thus, the total content of Fe-EDDHA in the soils was presumably enhanced by repeated applications. These experiments provided initial evidence of the effectiveness of Fe-DMA application in overcoming Fe deficiency in rice grown in alkaline soils. However, the stability of DMA in soil requires improvement and the expense of DMA synthesis precludes its practical use as a fertilizer.

**Synthesis and characterization of PDMA, a more stable and less expensive DMA analog**. DMA is costly to synthesize because it requires large amounts of expensive L-azetidine-2-carboxylic acid as the starting material[24,30–34], and the instability of DMA is likely caused by the highly strained four-membered ring of azetidine-2-carboxylic acid. To address these

critical issues, we investigated the synthesis and use of alternatives to L-azetidine-2-carboxylic acid, by transitioning from a four-membered azetidine ring to a five-membered ring with the aim of reducing molecular strain and improving the resistance to biodegradation. Five-membered analogs of DMA were synthesized using the procedure employed in our previous synthesis of DMA[24] from L-proline (Supplementary Fig 1 and Supplementary Note). In addition, to determine the necessity of the cyclic structure, acyclic analogs were synthesized using glycine and *N*-methyl glycine, the latter of which retains the tertiary amine moiety of DMA (Supplementary Figs. 1 and 2 and Supplementary Note). The analogs of L-proline, glycine, and *N*-methyl glycine were named PDMA, GDMA, and MGDMA, respectively. These synthetic analogs were confirmed to form a 1:1 complex with Fe(III) using high-resolution mass spectrometry analysis (Supplementary Figs. 3–5).

We evaluated the Fe transport activity of the analogs of DMA (Fig. 3a) using the barley transporter HvYS1, because of its high substrate selectivity[35]. Solutions (7.5 mM) of complexes of PDMA, GDMA, and MGDMA with FeCl$_3$ consisting of radioactive $^{55}$Fe (1/5 cold) in MES/Tris buffer (pH 6.0) were applied to Sf9 insect cells overexpressing HvYS1. After the cells and solutions had been mixed gently for 1 h and then washed, the radioactivity of $^{55}$Fe absorbed by the cells was measured using a scintillation counter. The intracellular radiation dose of cells treated with $^{55}$Fe-PDMA was identical to that of cells treated with $^{55}$Fe-DMA, but the intracellular radiation doses of cells treated with $^{55}$Fe-GDMA and $^{55}$Fe-MGDMA were considerably lower (Fig. 3b). Thus, the Fe-PDMA and Fe-DMA complexes were both transported by HvYS1. The cyclic structure was confirmed to be important for Fe transport activity, as confirmed by the six-membered analog (vide infra). Next, the abilities of transporters of other Poaceae species, such as maize (ZmYS1) and rice (OsYSL15), to transport PDMA were assayed by two-electrode voltage clamp analysis using *Xenopus laevis* oocytes. Fe-PDMA and Fe-DMA complexes were transported by HvYS1, ZmYS1, and OsYSL15 (Fig. 3c), implying that the transporters of a variety of Poaceae species are capable of taking up PDMA. In addition, in vivo the uptake of PDMA was confirmed by liquid chromatography coupled with tandem mass spectrometry (LC/MS) analysis of the xylem sap of rice plants supplied with Fe-PDMA from their roots, but PDMA was not detected in plants supplied with Fe-DMA (Fig. 3d). In contrast, DMA was detected in the xylem sap of rice plants supplied with both Fe-DMA and Fe-PDMA, because DMA is biosynthesized in rice (Supplementary Fig. 6). These results demonstrate that the Fe-PDMA complex was absorbed by the roots of Poaceae plants via YS1/YSL transporters, similar to the mechanism by which natural Fe-MAs are absorbed.

Moreover, qPCR analysis showed that Fe-PDMA and Fe-DMA exerted similar effects on the expression of Fe-deficiency-induced genes in rice grown in hydroponic culture (Supplementary Fig. 7). The transcript levels of *OsNAS2* (involved in DMA synthesis[36]) and of *OsIRO2*, (encoding a transcription factor induced by Fe-deficiency[37]) were markedly lower after application of Fe-DMA and Fe-PDMA compared with Fe-EDTA, implying that Fe-PDMA is as suitable for absorbing Fe as Fe-DMA, and is more suitable than Fe-EDTA. Therefore, the mode of action of PDMA is presumably similar to that of DMA[22].

Finally, the ability of PDMA to form complexes with Fe was investigated. Using Bjerrum's method[38], the constant stability of PDMA for Fe(III) (log K$_{Fe}^{III}$) was determined to be 17.1 (see "Methods" section), which is slightly lower than the previously reported stability of DMA (log K$_{Fe}^{III}$ = 18.4)[39,40]. These results indicate that PDMA is an inexpensive derivative of DMA with similar functionality.

**Effects of PDMA application to calcareous soil.** Based on the demonstrated Fe-transport activity of PDMA, we conducted a pot experiment using calcareous soil to compare the efficacy and stability of PDMA with those of DMA and EDDHA. To this end, we applied Fe-PDMA, Fe-DMA, and Fe-EDDHA to soil on one occasion at 4 days after transplantation. The rice plants already showed Fe deficiency at this point, and we evaluated their recoveries following the addition of chelates.

The SPAD values of rice plants supplied with Fe-PDMA and Fe-DMA increased to approximately 30 within a few days of application (Fig. 4a), at which point the effect of Fe-DMA remained, similar to the first experiment (Fig. 2a, b). Importantly, the SPAD values of leaves under the Fe-PDMA treatment increased to 40, a typical peak value of the darkest green rice leaves, at 12 days after application. Unlike the positive effect of continuous application of Fe-EDDHA (Fig. 2a), the rice plants subjected to one application of Fe-EDDHA showed an initial reduction in their SPAD values with no subsequent notable enhancement. This demonstrated the superior growth effects of Fe-PDMA and Fe-DMA than Fe-EDDHA. Moreover, elevated SPAD values of rice treated with Fe-PDMA persisted for about approximately 2 weeks (Fig. 4a, b), implying that Fe-PDMA was more stable than Fe-DMA in calcareous soil, as expected. A biodegradation assay showed that, compared with citrate, PDMA was much less rapidly biodegraded by microorganisms, likely due to the composition of the microbial community. In contrast, no biodegradation of EDTA was observed (Supplementary Table 1).

**Effects of the application of metal-free PDMA to calcareous soil.** Because natural phytosiderophores are secreted into rhizospheres and make complexes with Fe in the soil, we investigated whether application of unchelated PDMA to calcareous soil is effective for rice plants. Because sufficient Fe deficiency had not developed after 6 days, we evaluated the prevention of Fe deficiency by applications of chelates. As expected, the application of 30 μM unchelated PDMA to calcareous soil prevented Fe-deficiency chlorosis, as did PDMA applications both with Fe(III) and Fe(II) (Fig. 5a, b). In contrast, application of disodium EDTA did not prevent Fe-deficiency chlorosis. Measurement of Fe concentrations in soil solutions after incubation of calcareous soil with chelating agents revealed that PDMA solubilized Fe in soil (Supplementary Table 2), which showed that PDMA chelates native Fe in calcareous soil. The resulting Fe-PDMA complex was absorbed by the rice plants. It should be noted that there has never been the example of an artificial chelator, that prevents Fe deficiency without the addition of an Fe source until the development of PDMA in this study.

Although Fe concentrations were lower in the soil solution supplied with metal-free 30 μM PDMA than in the soil solution supplied with 30 μM Fe-EDDHA (Supplemental Table 2), the growth improvement by application of metal-free 30 μM PDMA was superior to that resulting from application of 30 μM Fe-EDDHA. Moreover, application of smaller amounts of metal-free PDMA remained effective in alleviating Fe deficiency chlorosis (Fig. 5a, b). The effects of PDMA application (1 or 3 μM) were similar to those of 30 μM Fe–EDDHA, suggesting that the effects of PDMA are more than 10-fold greater than those of Fe-EDDHA.

**Analysis of micronutrients taken into rice plants by PDMA.** MAs reportedly chelate Fe as well as other micronutrients[41] (e.g., Zn, Mn, and Cu). They also reportedly contribute to Zn absorption[42]. Consistent with the prior findings, our soil-incubation test revealed that PDMA solubilized Fe, Zn, and Cu in the soil both with and without fertilizer containing micronutrients (Supplementary Table 2). We next analyzed whether application of the chelates enhanced Fe concentrations in rice

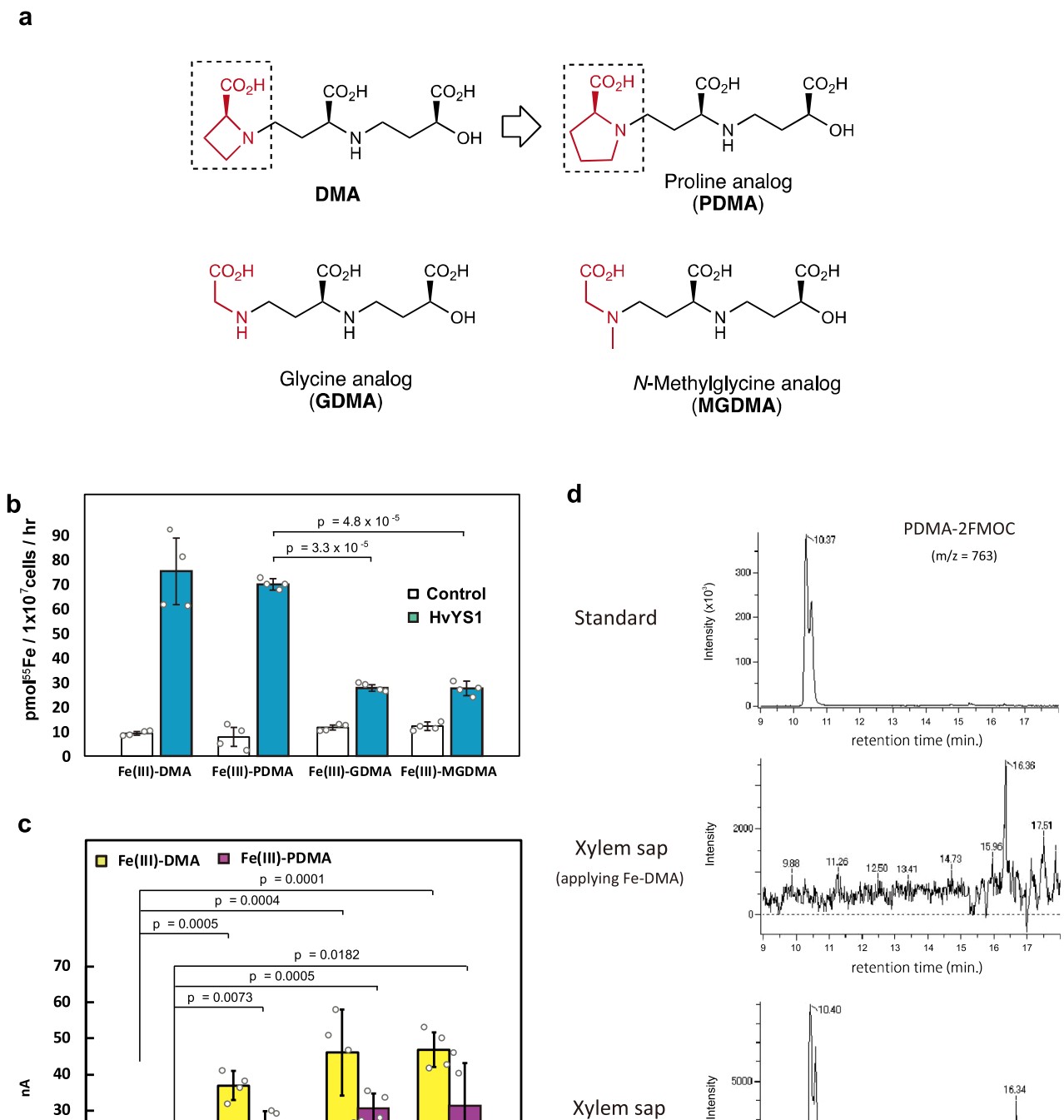

**Fig. 3 Transport of synthetic DMA analogs. b** Structures of synthetic DMA analogs. **b** Addition of 50 µM $^{55}$Fe(III)-DMA and $^{55}$Fe(III)-proline deoxymugineic acid (PDMA) to HvYS1-expressing Sf9 insect cells (1 × 10$^7$) enhanced their Fe(III)-transport activity. In contrast, $^{55}$Fe(III)-GDMA and $^{55}$Fe (III)-MGDMA only marginally enhanced the Fe(III)-transport activities of the cells. Each Fe(III) complex contained 10 mol% radioactive $^{55}$Fe. Sf9 insect cells transfected with empty vector were used as a control. Values are means with standard deviations of the four cell lysate replications (two-tailed *t*-test for Fe(III)-PDMA in HvYS1). **c** Fe(III)-transporting activities of synthetic DMA and PDMA in *Xenopus* oocytes injected with *HvYS1*, *OsYSL15*, or *ZmYS1* cRNA, or water (negative control) were measured by two-electrode voltage clamp analysis. Values are means with standard deviations of the four replicates (two-tailed *t*-test for water injection). **d** Detection of PDMA derivatized using 9-fluorenylmethoxycarbonyl chloride (FMOC) by liquid chromatography-time-of-flight mass spectrometry (LC-TOF-MS) analysis in the xylem sap of rice plants supplied with Fe-DMA and Fe-PDMA in the nutrient solution. DMA 2'-deoxymugineic acid, PDMA proline-2'-deoxymugineic acid, GDMA glycine-2'-deoxymugineic acid, MGDMA *N*-methylglycine-2'-deoxymugineic acid, HvYS1 *Hordeum Vulgare* yellow stripe 1, OsYSL15 *Oryza Sativa* yellow stripe 1 like transporter 15, ZmYS1 *Zea mays* yellow stripe 1.

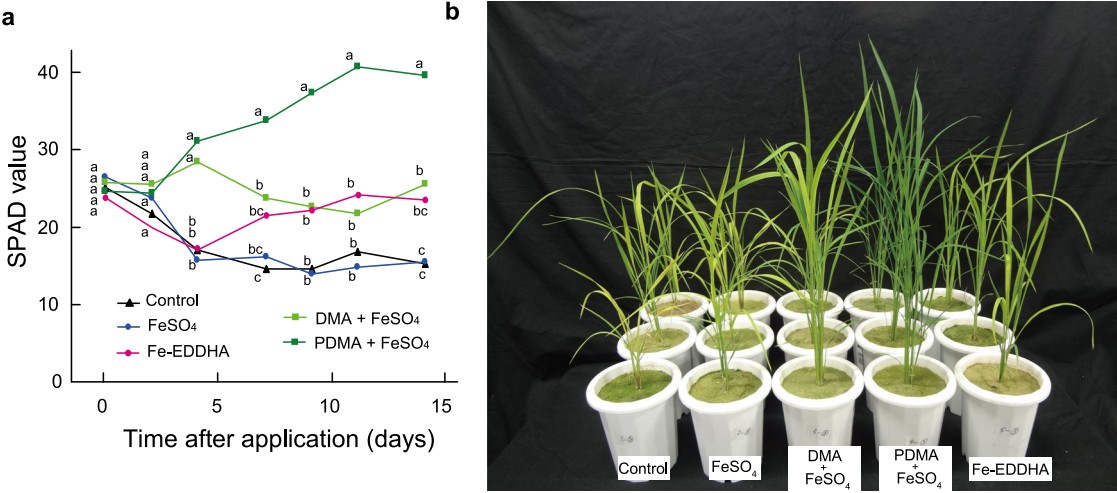

**Fig. 4 Effects of PDMA, a synthetic DMA derivative, on rice in calcareous soil. a** SPAD values of the newest leaves after a single application of the Fe chelates at 4 days after transplantation. DMA and PDMA were mixed with equimolar $FeSO_4$ in water before application. Values are the means of three replicates in which each replicate is average of three plants in each pot. Different letters indicate significant differences at $P < 0.05$ by Tukey's honestly significant difference test (two-sided). $P$-values for one-way analysis of variance (ANOVA) are follows: 0 days, 0.9394; 2 days, 0.2530; 4 days, 0.0008; 7 days, <0.0002; 9 days, <0.0001; 11 days, 0.0002; 14 days, <0.0001. **b** Rice plants at 14 days after a single application of the Fe chelates. DMA 2′-deoxymugineic acid, PDMA proline-2′-deoxymugineic acid, EDDHA ethylenediamine-$N,N'$-bis(2-hydroxyphenylacetic acid), SPAD soil and plant analyzer development.

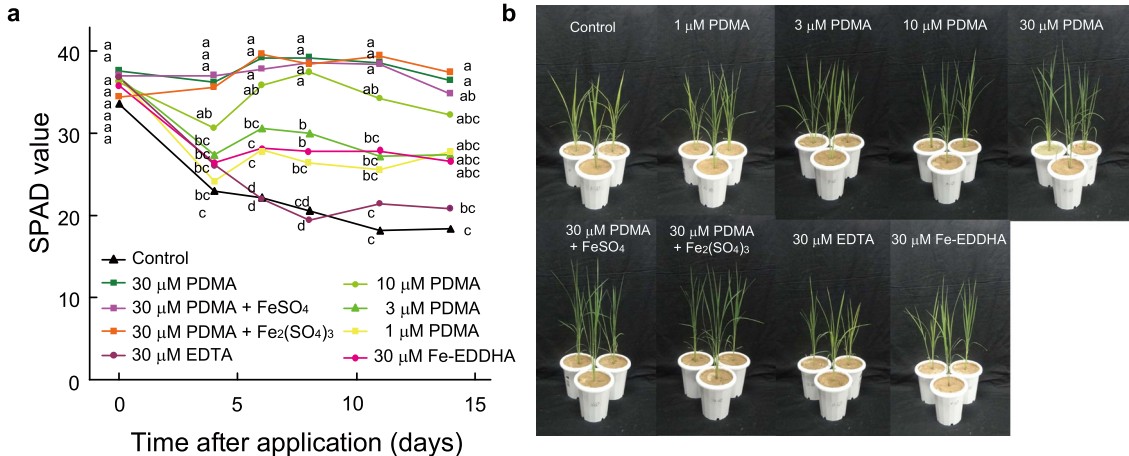

**Fig. 5 Effects of unchelated PDMA on rice in calcareous soil. a** SPAD values of the newest leaves after a single application of 1–30 µM unchelated PDMA compared with other Fe chelates at 6 days after transplantation. PDMA was also mixed with equimolar $FeSO_4$ or $Fe_2(SO_4)_3$. Values are the means of three replicates in which each replicate is average of three plants in each pot. Different letters indicate significant differences at $P < 0.05$ by Tukey's honestly significant difference test (two-sided). $P$-values for one-way analysis of variance (ANOVA) are follows: 0 days, 0.5287; 4 days, <0.0001; 6 days, <0.0001; 8 days, <0.0001; 11 days, <0.0001; 14 days, 0.0017. **b** Rice plants at 14 days after a single application of the Fe chelates. PDMA proline-2′-deoxymugineic acid, EDDHA ethylenediamine-$N,N'$-bis(2-hydroxyphenylacetic acid), EDTA ethylenediaminetetraacetic acid, SPAD soil and plant analyzer development.

plants by measuring Fe, Zn, Mn, and Cu concentrations in the newest leaves by inductively coupled plasma optical emission spectrometry. Metal concentrations in leaves were measured at 7 days after treatment to focus on the primary effects of the applications. Soil application of metal-free PDMA enhanced both the Fe concentrations of leaves and SPAD values in a manner similar to that of Fe(III)-PDMA (Fig. 6a, b), demonstrating that sufficient Fe was taken up for chlorophyll synthesis and storage in leaves. Soil application of Fe-EDDHA slightly enhanced SPAD values, but did not raise Fe concentrations in leaves. Presumably, small amount of Fe taken up by Fe-EDDHA was used preferentially in chlorophyll synthesis in the Fe-deficient condition. Soil applications of Zn-PDMA and Zn-EDTA enhanced Zn concentrations (Fig. 6c), although only Zn-PDMA treatment

enhanced the SPAD values (Fig. 6a). This treatment also enhance Fe concentrations, compared with control, Fe-EDDHA and Fe-EDTA treatments. The findings imply that a proportion of the supplied Zn-PDMA was converted to Fe-PDMA in the soil because of the high Fe content in the soil, and thus the rice plants absorbed both Zn-PDMA and Fe-PDMA. Mn concentrations were reduced by applications of metal-free PDMA, Fe-PDMA, Zn-PDMA, and Zn-EDTA. The greatest effect was observed for Zn-PDMA (Fig. 6d). Soil application of metal-free PDMA tended to enhance Cu concentrations (Fig. 6e), consistent with the soil-incubation test (Supplementary Table 2).

These data indicate that SPAD values were related to Fe concentrations rather than other micronutrients, and that PDMA may be useful for supplying Fe, as well as Zn.

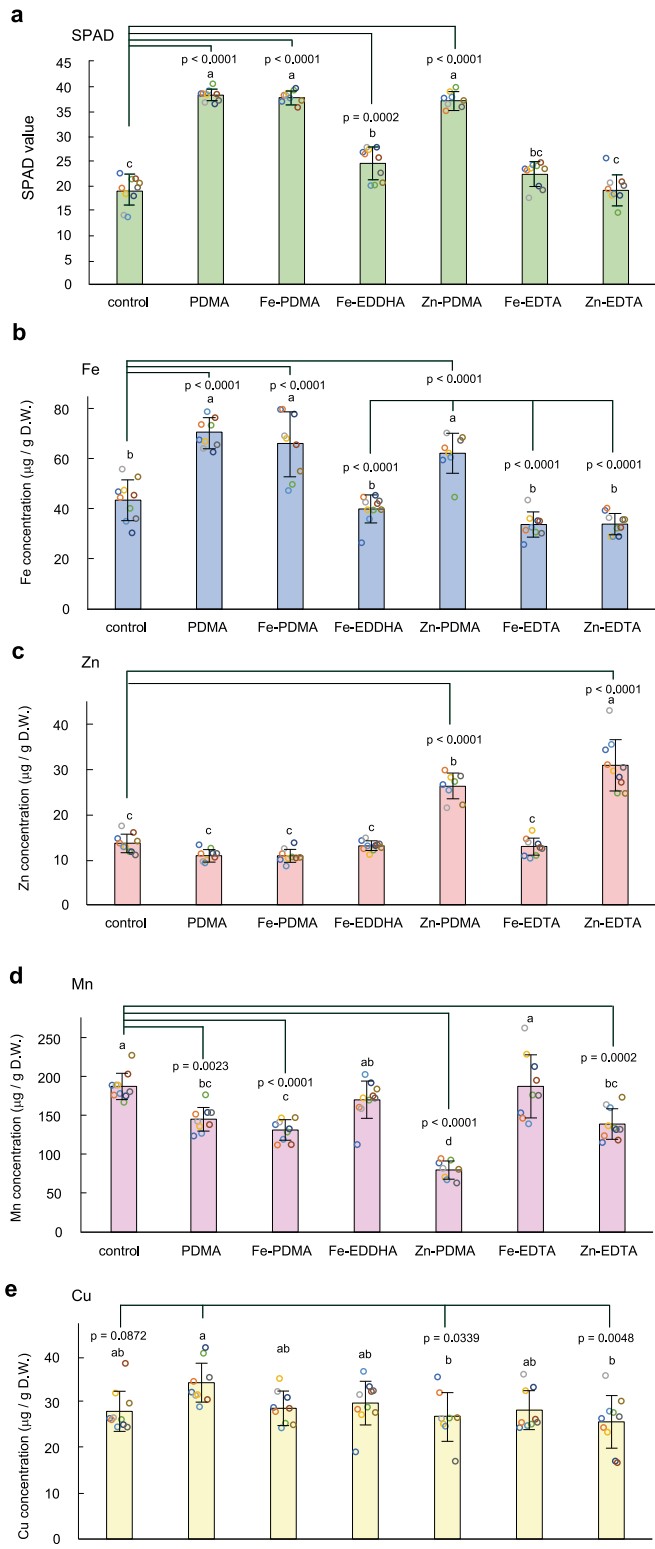

**Fig. 6 SPAD values and metal concentrations in the newest leaves of rice plants supplied with chelating agents. a** SPAD values were measured 7 days after a single application at 4 days after transplantation. **b**–**e** Metal concentrations in the newest leaves. Ten pots were used in each treatment, and three rice plants were grown in each pot. The three newest leaves in each pot were collected and used for measurement of SPAD values and metal concentrations by inductively coupled plasma optical emission spectrometry. The colors of the plots of each treatment through **a**–**e** indicate samples from the same pots. Values are means with standard deviations indicated by error bars ($n = 10$ for control, Fe-EDDHA and Zn-EDTA, $n = 9$ for PDMA, Fe-PDMA, and Fe-EDTA, and $n = 8$ for Zn-PDMA, after excluding outliers of Fe concentration), in which each replicate is either average of three plants in each pot for **a**, or total metal concentration of the three plants in each pot for **b**–**e**. Different letters indicate significant differences at $P < 0.05$ by Tukey's honestly significant difference test (two-sided). PDMA proline-2′-deoxymugineic acid, EDDHA ethylenediamine-N,N′-bis(2-hydroxyphenylacetic acid), EDTA ethylenediaminetetraacetic acid, SPAD soil and plant analyzer development, D.W. dry weight.

**Fig. 7 Synthesis of PDMA.** PDMA·HCl salt **7** was synthesized from allylglycine **1** with an overall yield of 38%. Purification by column chromatography was performed once. Boc *tert*-butoxycarbonyl, $^t$Bu*tert*-butyl, NaBH(OAc)$_3$ sodium triacetoxyborohydride, NaBH$_3$CN sodium cyanoborohydride, AcOH acetic acid.

**Analogs possessing other amino acids.** Because the amino acid position of L-azetidine-2-carboxylic acid was important for Fe transport activity, we examined additional DMA analogs with other amino acids. Specifically, we evaluated the effects of analogs of L-pipecolinic acid (six-membered ring) (PiDMA), 5-aminovaleric acid (long acyclic chain) (AvDMA), β-alanine (short acyclic chain) (ADMA), and L-proline (PDMA; positive control) on rice growth in calcareous soil. The six-membered analog promoted recovery from Fe deficiency with an effect similar to that of PDMA. Its efficacy was considerably greater than that of the acyclic analogs, supporting the functional importance of the cyclic structure (Supplementary Fig. 8). However, we focused on the five-membered analog due to the greater availability of L-proline, which is commonly produced industrially, thereby reducing the cost of the starting material. (Supplementary Fig. 8).

**Large-scale chemical synthesis of PDMA.** Following the verification of PDMA as a potent fertilizer for overcoming Fe deficiency in Poaceae species grown in alkaline soil, we investigated its efficacy in the field. To supply PDMA for a pilot field experiment, we improved its synthesis for larger scale applications (Fig. 7). The improved synthesis began with *N-tert*-butoxycarbonyl(Boc)-L-allylglycine *tert*-butyl ester **1**, which is readily available using Maruoka's catalysts[43]. One-pot ozone oxidation of **1** in methanol followed by a reductive amination reaction with 2.0 equiv of free L-proline **2** afforded coupling product **3**, together with a small amount of *N*-methyl-L-proline derived from the reaction with formaldehyde. The *N*-methyl-L-proline and other contaminants, including any excess L-proline, were readily removed based on their acid–base distribution, and the crude **3** was sufficiently clean for further reaction (see the crude $^1$H NMR chart of **3** in the Supplementary Fig. 43). Treatment of **3** with anhydrous hydrogen chloride in ethanol at 50 °C yielded diethyl ester **4** as a hydrogen chloride salt. Although the hydrogen chloride salt could be used directly for the subsequent reductive amination with **5** after evaporation of the reaction mixture as in the synthesis of DMA, reverse acid–base distribution for desalination was more suitable for large-scale synthesis due to the simplicity of pH adjustment for the next reaction. The method for preparing aldehyde **5** was also improved for large-scale synthesis, and **5** was readily obtained from L-malic acid on the scale of dozens of grams (see Supplementary Note). The reductive amination of crude free amine **5** with 1.1 equiv of aldehyde **5** in the presence of 50 equiv of acetic acid proceeded smoothly to yield protected PDMA **6** with no noticeable by-products (see the crude and purified $^1$H NMR charts of **6** in the Supplementary Figs. 45 and 46). After one-time silica gel column

chromatography throughout all synthetic schemes of PDMA, **6** was hydrolyzed by 2.2 equiv NaOH in aqueous solution (0.2 M) at 0 °C followed by 1 M HCl at room temperature to yield PDMA as HCl salt **7** (PDMA-HCl), including 2.2 equiv NaCl. Although the crude PDMA-HCl could be purified by demineralization using an ion-exchange resin, HCl salt **7** was confirmed to have an efficacy similar to that of desalinated PDMA in the rice growth experiment in calcareous soil (Supplementary Fig. 9). Therefore, PDMA production for field experiments was adopted to supply the product as an HCl salt, which avoided the cumbersome desalination process using ion-exchange resin. Ultimately, we demonstrated practical scalable synthesis of PDMA, 42.5 g of **7** along with 11.9 g of NaCl, from 77.8 g of **1** in our laboratory. Because of the limited availability of MAs (DMA is approximately $1000/mg), the development of PDMA will dramatically advance research involving MAs by resolving the supply issue that has been a persistent bottleneck.

**Efficacy of PDMA in calcareous soil in the field.** We conducted a pilot field experiment using rice grown in calcareous soil placed in blocks (1 m × 1 m × 0.5 m depth) covered with vinyl sheeting, into which 35-day-old rice seedlings were transplanted. Unlike the above pot experiment, it was difficult to consistently control the water content of the soil in each block. The water level was maintained above the soil surface by the frequent addition of tap water. The plants developed Fe-deficiency chlorosis 14 days later, at which point metal-free PDMA, Fe-DTPA, and Fe-EDDHA were applied to the soil. The effect of 30 μM PDMA as a soil solution was apparent at 1 week after application, as indicated by the SPAD values (Fig. 8a). Notably, this benefit was evident for 4 weeks after application (Fig. 8b). In contrast, the 3 μM PDMA and 30 μM Fe-EDDHA treatments overcame the Fe deficiency chlorosis after 2 weeks, but the effects were weaker. Fe-DTPA was ineffective for alleviating Fe deficiency chlorosis within that timeframe. Another greenhouse pot experiment using calcareous soil confirmed that 30 μM metal-free PDMA was more effective than 3 μM PDMA (Supplementary Fig. 9). The growth test in the pilot field was continued for 4 weeks during the maintenance of high oxide-redox potential in soil (Supplementary Fig. 10). Subsequently, the condition of the test field became inappropriate for Fe-fertilizer evaluation after the soil redox potential decreased with the prolonged water-logged period, which enhanced the level of ferrous ion in the soil[12]. Thus, the effect of PDMA for enhancing the Fe contents of the grain could not be clarified because of the other factors related to the reduced state of the soil. For recovery from Fe deficiency, the superiority of PDMA was also observed in an outside pilot-field with the development of large-scale synthesis of PDMA.

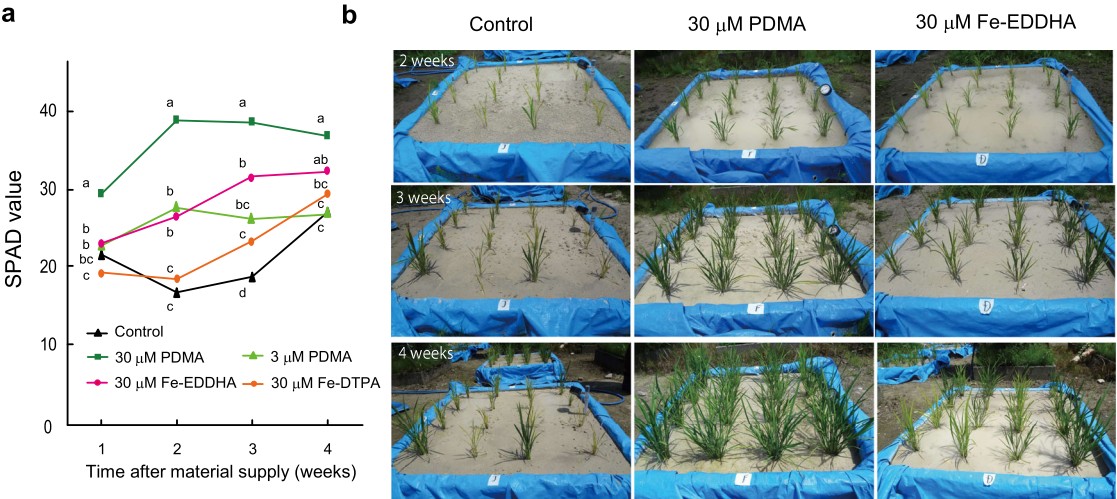

**Fig. 8 Pilot field experiment showing the effects of chelates on rice growth in calcareous soil. a** SPAD values of the newest leaves from 1 to 4 weeks after chelate application, showing the superiority of 30 μM unchelated PDMA and 30 μM Fe-ethylenediamine di(o-hydroxyphenylacetic) acid (EDDHA) in overcoming Fe deficiency. Values are the mean of three replicates in which each replicates is control, 48 (Contol), 32 (30 μM PDMA, 3 μM PDMA, 30 μM Fe-EDDHA), and 16 (30 μM Fe-DTPA). Different letters indicate significant differences at $P < 0.05$ by Tukey's honestly significant difference test (two-sided). $P$-values for one-way analysis of variance (ANOVA) are follows: 1 week <0.0001, 2 weeks <0.0001, 3 weeks <0.0001, 4 weeks <0.0001. **b** Photographs of rice plants at 2–4 weeks after treatment. PDMA proline-2′-deoxymugineic acid, EDDHA ethylenediamine-N,N′-bis(2-hydroxyphenylacetic acid), DTPA diethylenetriaminepentaacetic acid.

## Discussion

Soils unfavorable for agriculture cover two-thirds of land globally, more than half of which is categorized as alkaline/calcareous soil[1,2]. Thus, the development of agriculture in alkaline/calcareous soils is presumed to offer as a practical means of enhancing food production without causing severe environmental destruction. Because Fe deficiency is one of the agricultural problems in alkaline/calcareous soil, the development and application of Fe-chelate have been the subject of intensive research in soil science and plant nutrition. However, practical use of Fe-chelates is not fully established, despite the reported utilization of some natural and synthetic phytosiderophores.

In this study, PDMA showed considerable potential for enabling agriculture in alkaline/calcareous soil. This innovative fertilizer was derived from the natural phytosiderophore DMA, to overcome its instability and high synthesis cost. Compared with conventional synthetic artificial Fe chelates, the natural chelate-based PDMA has several advantages. First, direct absorption of PDMA into plant roots is simpler and more effective, compared with conventional synthetic chelates that require replacement of Fe ligands by secreted MAs, thus enabling PDMA to promote recovery from Fe deficiency in calcareous soil. Indeed, the effect of PDMA was significantly superior to that of Fe–EDDHA (Figs. 4a and 5a), a conventional chelate that is considered effective in calcareous soils. Second, although artificial Fe chelates tend to persist in soil, PDMA was biodegradable by soil microorganisms (Supplementary Table 1), indicating that environmental destruction is unlikely to be an additional concern. Third, PDMA retains the functions of natural MAs, such that unchelated PDMA forms a complex with insoluble Fe(III) in alkaline/calcareous soils, and the resulting Fe-PDMA complex is absorbed by roots (Figs. 5a and 6). Thus, external application of Fe is not required for plant recovery from Fe deficiency. To our knowledge, there has never been chelating agent that does not require the addition of Fe until this study.

The facilitation of agriculture in pilot fields with calcareous soil implies the utility of PDMA as a fertilizer. Moreover, the synthesis of a large quantity of PDMA in our laboratory indicates the practicality of industrial-scale production for agricultural use, although its synthesis requires further improvement. Practical application of PDMA also requires its safety to be demonstrated. We have begun a safety investigation, the preliminary results of which show that PDMA does not exert any cytotoxic effects (Supplementary Fig. 11). Thus, we presume that practical application of PDMA will become a reality in the near future.

We found that in Poaceae species, PDMA is transported by transporters such as ZmYS1 in maize, HvYS1 in barley, and OsYSL15 in rice (Fig. 3c). In this study, the superiority of PDMA was demonstrated in rice, a Poaceae member that is highly sensitive to Fe-deficiency stress. Therefore, PDMA is expected to be effective in all other Poaceae plants. In addition, PDMA has the potential for application to non-Poaceae plants, which can use MAs secreted from Poaceae plants via intercropping[18,19,44]. Another transporter of MAs, AhYSL1, is localized to the epidermis of peanut roots and transports Fe-DMA[45], implying that MAs and PDMA can be externally applied. Thus, PDMA is likely to be effective in all plant species, not only Poaceae. Zn deficiency is another major issue associate with crop loss in calcareous soil. Zn-MAs complexes are also considered as the key factor for the acquisition of Zn in Poaceae plants[42], and their contribution is also important for Zn translocation within a plant[46]. PDMA may resolve Zn deficiency and has the potential to be used as a universal fertilizer for micronutrients.

Finally, the development of PDMA appears to involve simply replacing the four-membered ring by a five-membered ring. However, this approach has not been attempted during the 40-year interval since the discovery of MAs[3], primarily because practical application of MAs as a fertilizer to alkaline soils has been considered as impossible due to their high cost of synthesis and limited availability from natural sources. Thus, the application of other synthetic Fe chelates to soil has been studied intensively. Here, we rediscovered the power of MAs in that natural DMA is an effective fertilizer in calcareous soil. We subsequently developed practical PDMA. Despite our simple strategy, the development of PDMA offers up the possibility of greening infertile alkaline soils.

## Methods

### Chemical synthesis and compound characterization.
Detailed synthetic procedures and compound characterization are described in the Supplementary Note, along with [1]H and [13]C NMR spectra were also available.

### Mass spectrometric analysis.
Mass spectra were obtained using an LTQ Orbitrap Elite (Thermo Fisher Scientific, Bremen, Germany) with an electrospray ionization (ESI) ion source. MS analysis was performed in negative ion mode. The ESI-MS operating conditions were as follows. ESI voltage [kV]: $-3.5$; capillary temperature [°C], 275; sheath gas flow rate [arb], 8; Mass scan range: $m/z$ 150–1000; and resolution, 60,000. Acetonitrile/water (1:1; v/v) solution containing PDMA, GDMA, or MGDMA (50 μM) and each Fe(III)-complex (50 μM) were prepared for MS measurements. ESI-MS analysis injected these solutions at 5 μL/min flow rate.

### Rice growth in pots of calcareous soil
*Experiment 1 (shown in Fig. 2).* Rice (*Oryza sativa* L. cv Nipponbare) seeds were soaked in water for 7 days to induce germination. Three seedlings grown similarly were transplanted into 600 mL pots filled with 800 g calcareous soil (pH 9; Nihonkai Hiryou Co. Ltd., Takaoka, Toyama, Japan), as was analyzed previously[25]. The pots contained 2-g controlled-release (for 70 days) NPK-micronutrient-type fertilizer (Longtotal70–313; JCAM Agri. Co., Ltd., Tokyo, Japan) with the following composition (in g/kg): N 130, $P_2O_5$ 110, $K_2O$ 130, MgO20, MnO 1,$B_2O_3$ 0.6, Fe 5 (as Fe-EDTA), Cu 0.7, and Zn 0.3). A volume of 270 mL distilled water was supplied to reach saturation. The bottom of the pot was covered with paper so that water could be drained. Solutions of five Fe chelates were prepared as mixtures of 0.2 mM $FeSO_4$ and either DMA (20), citrate (Cas No. 77-92-9; Wako Co. Ltd., Tokyo, Japan), or deferoxamine mesylate (Cas No. 138-14-7); Merck KgaA, Darmstadt, Germany) along with 0.2 mM Fe as Fe-EDTA (Cas No. 15708-41-5; Dojindo Molecular Technologies, Inc., Kumamoto, Japan) and Fe–EDDHA (6% Fe (3% o, o-EDDHA) CresCal, Aglukon Spezialduenger GmbH & Co. KG, Dusseldorf, Germany). The 0.2 mM solution of Fe and chelates was stirred for 1 h at room temperature before application. The Fe chelate solutions were applied as 8 μmol amounts at 3, 5, 7, 10, 12, and 14 days after transplantation.

The efficacy of the chelates in overcoming Fe deficiency was assessed using the SPAD values of the newest leaves first measured at 14 days after transplantation. Average value of three plants in each pot was used as each replicate. In addition, distilled water was applied to the soil three times weekly to maintain soil saturation (270 mL water/800 g soil). The plants were cultured in a growth chamber under a 14-h light/28 °C (200 μE/s m[2]) and 10-h dark/23 °C cycle. Three replicate pots were used for each condition in the growth experiment. Statistical analysis (Tukey's honestly significant difference test) was performed using JMP14 software (SAS Institute, Cary, NC, USA).

*Experiment 2 (shown in Fig. 4a, b).* Plants were grown in pots as in Experiment 1 with the following modifications: after 7 days of soaking to induce germination, the seedlings were transferred onto a Saran net floating on distilled water for 3 days, and then transplanted to calcareous soil. Solutions of $FeSO_4$, a mixture of $FeSO_4$ and DMA, PDMA, or Fe–EDDHA were applied to the soil at 4 days after transplantation at which point the SPAD values were first measured. Three replicate pots were used for each condition in the growth experiment.

*Experiment 3 (shown in Fig. 5a, b).* Plants were grown in pots as in Experiment 2, with the following modifications: after 7 days of soaking to induce germination, the seedlings were transferred onto a Saran net floating on nutrient solution[42] (0.7 mM $K_2SO_4$, 0.1 mM KCl, 0.1 mM $KH_2PO_4$, 2.0 mM $Ca(NO_3)_2$, 0.5 mM $MgSO_4$, 10 μM $H_3BO_3$, 0.5 μM $MnSO_4$, 0.2 μM $CuSO_4$, 0.5 μM $ZnSO_4$, 0.05 μM $Na_2MoO_4$, and 0.1 mM Fe-EDTA) for 7 days, and then transplanted to calcareous soil. Six days after transplantation, solutions of Fe as $Fe_2(SO_4)_3$ or $FeSO_4$ with PDMA, or Fe-EDDHA were applied to the soil at a rate of 30 μM, and unchelated PDMA solution was applied to the soil at rates of 1, 3, 10, or 30 μM. The SPAD values were first measured at 6 days after transplantation on the same day as chelate addition. Three replicate pots were used for each condition in the growth experiment.

*Experiment 4 (shown in Supplementary Fig. 8).* Plants were grown in pots as in Experiment 1 with the following modifications: the solutions of Fe sulfate or a mixture of Fe sulfate and chelating agent (five-membered ring analog equal to PDMA, six-membered ring analog), or Fe-EDDHA were applied to the soil on five occasions: 6, 8, 11, 13, and 15 days after transplantation. The SPAD values were first measured at 11 days after transplanting. Two replicate pots were included for each condition in the growth experiment.

*Experiment 5 (shown in Supplementary Fig. 9).* Rice plants (cv. Koshihikari) were germinated and grown in nursery soil in a growth chamber. Seedlings were transplanted after 23 days into 4 kg of calcareous soil containing 10 g controlled-release NPK-micronutrient-type fertilizer (as described above) in pots (159 φ × 190 mm) after nursery soil had been removed from the root surface. Water was added to the soil three times weekly to maintain soil saturation. The SPAD values were measured after transplantation. PDMA hydrochloride salt and 2.2 equiv. NaCl were applied to the soil as a 3 or 30 μM soil solution at 27 days after transplanting.

### Determination of metal concentrations in rice plants.
The rice plants were grown in a manner similar to that of Experiment 3 (described above), with the following alterations: after 7 days of soaking to induce germination, the seedlings were transferred onto a Saran net floating on nutrient solution similar to that used in experiment 3 for 6 days, then transplanted to calcareous soil. After 4 days, 30 μM chelate solutions of $Fe_2(SO_4)_3$ with PDMA, $ZnSO_4$ with PDMA, Fe–EDDHA, Fe-EDTA, $ZnSO_4$ with EDTA, unchelated PDMA or water control were applied to the soil. Three replicate pots were used for each condition. After 7 days, the three newest leaves per pot were cut and dried for 3 days at 70 °C, then wet-ashed with 1.5 ml of 4.4 M $HNO_3$ and 1.5 ml $H_2O_2$ for 20 min at 220 °C using a MarsXpress oven (CEM). The Fe, Zn, Mn, and Cu concentrations were measured as above.

### Pilot field experiment.
An experiment was conducted in a field at Ishikawa Prefectural University (Nonoichi, Ishikawa, Japan; 36.507°N, 136.597°E), in which approximately 0.5 *t* of calcareous soil was placed into blocks (internal dimensions: 1 m × 1 m × 0.5 m depth) contained within a tarpaulin (Arcland Sakamoto Co., Ltd, Niigata, Japan) to ensure that the calcareous soil did not mix with the surrounding soil of a different type. Thereafter, controlled-release (for 70 days) NPK-micronutrient-type fertilizer (80 g total; Longtotal70–391; JCAM Agri. Co., Ltd.; containing (in kg): N 130, $P_2O_5$ 90, $K_2O$ 110, MgO 20, MnO 1,$B_2O_3$ 0.6, Fe 5 as Fe-EDTA, Cu 0.7, and Zn 0.3) was mixed with the calcareous soil at a depth of 7–12 cm, and the same fertilizer (20 g) was applied at a depth of 2 cm. Tap water was added to saturate the soil into which 35-day-old rice (cv. Koshihikari) seedlings (~13 cm in height) with roots cleaned of nursery soil (pH 5) were transplanted. Three seedlings were transplanted from each hill and set in 3-cm-deep holes with 25-cm spacing in both the row and column directions, yielding 16 hills with transplants in each block. The water level was maintained above the soil surface by frequent addition of tap water. Treatments of 2.55 g of PDMA-HCl, 0.26 g of PDMA-HCl, 4.77 g of Fe-EDDHA (Fe 6% [3% $o,o$-EDDHA]), and 2.51 g of Fe-DTPA (Fe 11%; Aglukon Spezialduenger GmbH & Co. KG) were applied at 14 days after transplantation, and plant height (highest plant on each hill) and SPAD values (average of the newest leaves of the three plants on each hill) were measured weekly after the chelating agent application. The number of hills in each treatment was as follows: control, 48; unchelated PDMA 30 μM, 32; unchelated PDMA 3 μM, 32; Fe-EDDHA 30 μM, 32; and Fe-DTPA 30 μM, 16. Statistical analysis (Tukey's honestly significant difference test) was performed using JMP14 software (SAS Institute, Cary, NC, USA). Soil Redox Potential (Eh, mV) was measured at representative three points in the experimental blocks using Eh meter FV-702 (Fujiwara Scientific Company Co. Ltd., Tokyo, Japan).

### Measurement of PDMA-Fe(III) transport activity by heterologous expression of a DMA-Fe(III) transporter.
PDMA-Fe(III) transport activity was measured in insect cells heterologously expressing a DMA-Fe(III) transporter from barley (HvYS1; GenBank accession No. AB214183) (Supplementary Table 3). The gene encoding HvYS1 was PCR-amplified from cDNA and cloned into the pFastBac Dual expression vector (Thermo Fisher, Waltham, MA, USA), fused to an octa-histidine tag at the C-terminus. HvYS1 was expressed in baculovirus-infected Sf9 cells (Therm Fisher) using the Bac-to-Bac system in accordance with the manufacturer's instructions (Thermo Fisher).

DMA-Fe(III) complexes were prepared as follows. DMA, PDMA, GDMA, and MGDMA were dissolved in 10 mM MES/Tris buffer (pH 6.0) to a final concentration of 200 mM. Equal amounts of 180 mM $FeCl_3$ and 20 mM [55]Fe solution (NEZ043; 37.0 MBq, Perkin-Elmer, Waltham, MA, USA) were mixed to produce a 100 mM Fe solution. Next, equal amounts of Fe solution (100 mM) and each DMA solution (200 mM) were mixed and diluted in 10 mM MES/Tris buffer (pH 6.0) to a final concentration of 7.5 mM Fe-DMAs (Fe:DMAs = 1:2). To remove the undissolved fraction, the Fe-DMAs were subjected to centrifugal filtration (Merck Ultra Free-MC-GV).

Sf9 cells were cultured for 3 days after transfection with HvYS1 or empty vector in SF900II medium (Thermo Fisher), and 5 mL ($2 \times 10^6$ cells/mL) of cell were harvested by centrifugation at $1,300 \times g$ for 5 min. Next, the cells were resuspended in 1 mL of SF900II medium. An aliquot (7 μL) of each 7.5 mM Fe(III) complex containing 10 mol% [55]Fe was added to the cell suspensions in 1 mL medium (final concentration of Fe(III) complex was 50 μM), which were then gently mixed at room temperature for 1 h. The cells were harvested by centrifugation at $1700 \times g$ for 5 min. The harvested cells were washed three times by centrifugation at $1700 \times g$ for 5 min at 4 °C with 1.0 mL of phosphate-buffered saline to remove [55]Fe that had not been taken up by the cells[47]. The cells were then lysed in 300 μL of 1 M NaOH and 200 μL of phosphate-buffered saline by vortex mixing. Finally, the cell lysates were transferred to a counting vial (Wheaton, No. 986492) containing 3 mL of the cocktail (Ultima Gold, Perkin-Elmer). The radioactivity of [55]Fe was measured using a Tri-Carb 1900CA liquid scintillation analyzer (Packard Instruments, Downers Grove, IL, USA).

### Electrophysiological studies in *X. laevis* oocytes.
Late-stage oocytes (stage V–VI) are removed from the ovaries of female *Xenopus laevis* aged 1–2 years (Kato-S Science, Chiba, Japan). The *X. laevis* is anesthetized by gradually cooling the frog in an ice water bath at 4 °C for 30–45 min and harvested oocytes are defolliculated for about 110 min with 2 mg/ml collagenase (sigma-Aldrich) in

sterile OR2 solution containing: 96 mM NaCl, 2 mM KCl, 1 mM MgCl$_2$, and 5 mM HEPES, pH 7.4. The methods of electrophysiological studies in *X. laevis* oocytes expressing *OsYSL15* (rice) were the same as that for *HvYS1* (barley) and *ZmYS1* (maize, GenBank accession no. AF186234) (Supplementary Table 3)[35,48]. The open reading frame of *OsYSL15* (rice, GenBank accession no. AB190923) (Supplementary Table 3) was PCR-amplified from cDNA by PCR using the forward primer 5′-GCTCTAGACCACCATGGAGCACG−3′ and reverse primer 5′-CGCGGGAT CCTTAGCTTCCAGGCGTAAACTTC−3′. The PCR products of *HvYS1*, *ZmYS1*, and *OsYSL15* were purified using a QIAquick PCR Purification Kit (Qiagen, Hilden, Germany) and inserted into the *Xba*I and *Bam*HI sites of the pSP64 *X. laevis* oocyte expression vector (Promega, Madison, WI, USA). The *HvYS1*/pSP64 and *ZmYS1*/pSP64 vectors were linearized with *Bam*HI, and the *OsYSL15*/pSP64 vector was linearized with *Eco*RI. cRNA transcription was carried out using the SP6 mMESSAGE mMACHINE Kit (Ambion, Austin, TX, USA). *X. laevis* oocytes were injected with 50 nL each of *HvYS1*, *ZmYS1*, and *OsYSL15* cRNA and incubated in ND-96 medium (96 mM NaCl, 2 mM KCl, 1.8 mM CaCl$_2$, 1 mM MgCl$_2$, and 5 mM HEPES) for 72–96 h at 18 °C. *X. laevis* oocytes (Kato-S Science, Chiba, Japan) were voltage-clamped at −60 mV using an OC-725C oocyte clamp (Warner Instruments, Hamden, CT, USA) and placed in an open chamber with continuous perfusion of ND96 buffer (pH 6.0) with or without substrates. Steady-state currents were obtained after the addition of DMA-Fe(III) or PDMA-Fe(III) complexes at 200 μM in 10 mM MES/Tris buffer (pH 6.0). Data acquisition and analysis were performed using p-Clamp 10 (Molecular Devices, Sunnyvale, CA, USA). Oocytes injected with water were used as negative control.

**LC-TOF-MS of xylem sap of rice plants cultivated in nutrient solution**. Sixteen-day-old rice (cv Nipponbare) seedlings were cultivated hydroponically for 20 days in an Fe-containing nutrient solution similar to that used in experiment 3. Next, the seedlings were transferred to an Fe-depleted nutrient solution for 10 days. PDMA and DMA were mixed with Fe chloride at 500 μM overnight and added to the nutrient solution as 30 μM Fe. Subsequently, xylem sap was collected for 3 h and subjected to analysis by LC/ESI-TOF-MS, as following previous method[49]; 10 μL of 1 M sodium borate (pH 8.0) was added to 5 μL of xylem sap solution (or 0.24 mM DMA and 0.25 mM PDMA as standards) and then 10 μL of 50 mM EDTA (pH 8.0) was added. A 40 μL aliquot of 50 mM FMOC-Cl (Tokyo Chemical Industry) dissolved in CH$_3$CN was added to the solution. The reaction mixture was incubated at 60 °C for 15 min. The reaction was stopped by 5 μL of 5% formic acid. Authentic samples of DMA and PDMA (250 μM) were used as standards. LC-ESI-TOF-MS measurement was carried out using a JSM-T100LC AccuTOF (JEOL, Tokyo, Japan) in ESI+ mode. The desolvent temperature was 300 °C, the orifice 1 temperature was 120 °C and the ESI needle voltage was 2000 V. The LC separation was performed using a Synergi Hydro RP column (4 μm, 150 × 2.00 mm; Phenomenex, Torrance, CA, USA) with a solvent of 0.5% formic acid, 36% water and 63.5% CH$_3$CN.

**qPCR of roots of rice plants cultivated in nutrient solution**. Thirteen-day-old rice (cv. Nipponbare) seedlings were cultivated hydroponically for 7 days in an Fe-containing nutrient solution similar to that used in experiment 3. Next, the seedlings were transferred to an Fe-depleted nutrient solution for 5 days. PDMA, DMA, and EDTA were mixed with Fe chloride at 4 mM for 30 min and added to the nutrient solution as 30 μM Fe. After treatment for 4 days, roots were collected under liquid nitrogen, and total RNA was purified using an RNeasy Plant Mini Kit (Qiagen). First-strand cDNA was synthesized using ReverTra Ace qPCR RT Master Mix with gDNA Remover (Toyobo, Osaka, Japan). Real-time PCR was performed using a StepOnePlus™ Real-Time PCR System (Applied Biosystems, Foster City, CA, USA) with GoTaq qPCR Master Mix (Promega). The forward and reverse primers for *OsNAS2* were 5′-TGAGTGCGTGCATAGTAATCCTGGC-3′ and 5′-CAGACGGTCACAAACACCTCTTGC-3′, and those for *OsIRO2* were 5′-CAG-CATTTTGTGAAAGGTTGGAG-3′ and 5′-TTATTATCAGCTAACCAAATGC-TATATTTAAC-3′ (Supplementary Table 3). Transcript abundance was normalized to that of alpha-2 tubulin and was expressed as a ratio relative to that in the Fe-deficient plants by the $2^{-\Delta\Delta CT}$ method, in accordance with the manufacturer's instructions.

**Determination of stability constant**. Potentiometric titrations of solutions of (i) PDMA·2HCl (5 mM) and (ii) PDMA·2HCl (5 mM) + Fe$^{3+}$ (5 mM) were carried out at 20 °C at a constant ionic strength (0.1 M KNO$_3$). NaOH solution was used as the titrant. The pH of the solutions was measured using a digital pH meter with an accuracy of 0.01 pH units.

The proton dissociation constant of PDMA was determined by least-squares analysis of the PDMA·2HCl titration curve. The theoretical volume of NaOH required to reach a certain pH, $V_{Bcal}$, can be expressed as follows:

$$V_{Bcal} = \left\{ \frac{\frac{K_{a1}}{[H^+]} + 2\frac{K_{a1}K_{a2}}{[H^+]^2} + 3\frac{K_{a1}K_{a2}K_{a3}}{[H^+]^3} + 4\frac{K_{a1}K_{a2}K_{a3}K_{a4}}{[H^+]^4} + 5\frac{K_{a1}K_{a2}K_{a3}K_{a4}K_{a5}}{[H^+]^5}}{1 + \frac{K_{a1}}{[H^+]} + \frac{K_{a1}K_{a2}}{[H^+]^2} + \frac{K_{a1}K_{a2}K_{a3}}{[H^+]^3} + \frac{K_{a1}K_{a2}K_{a3}K_{a4}}{[H^+]^4} + \frac{K_{a1}K_{a2}K_{a3}K_{a4}K_{a5}}{[H^+]^5}} C_A \right. \\ \left. - ([H^+] - [OH^-]) \right\} \frac{V_A}{C_B + ([H^+] - [OH^-])}, \quad (1)$$

where $V_A$ is the volume of PDMA and $K_{a1}$, $K_{a2}$, $K_{a3}$, $K_{a4}$, and $K_{a5}$ are the first,

second, third, fourth, and fifth proton dissociation constants of PDMA, respectively. $C_A$ and $C_B$ are the concentrations of PDMA and NaOH, respectively. Using Microsoft Excel Solver (Microsoft Corp, Redmond, WA, USA), the proton dissociation constants of PDMA were computed by minimizing

$$\sum_{pH} \left( V_{Bcal} - V_{Bexp} \right)^2, \quad (2)$$

where $V_{Bexp}$ is the volume of NaOH added: p$K_{a1}$ = 2.01, p$K_{a2}$ = 2.47, p$K_{a3}$ = 3.12, p$K_{a4}$ = 7.63, and p$K_{a5}$ = 9.23.

The stability constant of the 1:1 PDMA-Fe$^{3+}$ complex was determined by the Bjerrum method[38]. The average number of ligands attached to the metal ion, $\bar{n}_L$, in the PDMA-Fe$^{3+}$ complex can be expressed as follows:

$$\bar{n}_L = \frac{\left( C_L - \frac{[H^+]^5}{K_{a1}K_{a2}K_{a3}K_{a4}K_{a5}} + \frac{[H^+]^4}{K_{a2}K_{a3}K_{a4}K_{a5}} + \frac{[H^+]^3}{K_{a3}K_{a4}K_{a5}} + \frac{[H^+]^2}{K_{a4}K_{a5}} + \frac{[H^+]}{K_{a5}} + 1 \right)[L]}{C_M}, \quad (3)$$

$$[L] = \frac{(5-f)C_L - [H^+] + [OH^-]}{\frac{5[H^+]^5}{K_{a1}K_{a2}K_{a3}K_{a4}K_{a5}} + \frac{4[H^+]^4}{K_{a2}K_{a3}K_{a4}K_{a5}} + \frac{3[H^+]^3}{K_{a3}K_{a4}K_{a5}} + \frac{2[H^+]^2}{K_{a4}K_{a5}} + \frac{[H^+]}{K_{a5}}}, \quad (4)$$

where $C_L$ and $C_M$ are the total concentrations of PDMA and Fe$^{3+}$, respectively, and [L] and $f$ are the concentration of free PDMA and the number of moles of base added per mole of ligand, respectively. In the Bjerrum method, the value of $-\log$ [L] at $\bar{n}_L$ = 0.5 produces the stability constant, log $K$. The stability constant of the 1:1 PDMA-Fe$^{3+}$ complex was estimated to be log $K$ = 17.1.

**Biodegradation assay**. The biodegradabilities of citrate, EDTA, and PDMA were assessed in Japan Food Research Laboratories (Tokyo, Japan), in accordance with the OECD Guideline for Testing of Chemicals (301A) (https://www.oecd.org/chemicalsafety/risk-assessment/1948209.pdf).

**Metal dissociation test of calcareous soil with chelating agent**. A 100-g aliquot of calcareous soil with 0.12 g of fertilizer (LongTotal70−313; described above) was mixed with 35 mL of 30 μM chelate solution (PDMA, EDTA·2Na, Fe-EDTA, Fe-EDDHA or no chelate) and incubated for 1 h at room temperature. Then, 70 mL of distilled water was added and the mixture was shaken for 1 h at room temperature. The supernatant was filtered through 5 C paper (ADVANTEC CO., LTD., Tokyo, Japan). The filtered solution was mixed with 240 μL of HNO$_3$ (final 2%), and the Fe, Zn, Mn, and Cu concentrations were measured by inductively coupled plasma optical emission spectrometry (ICPS-8100; Shimadzu, Kyoto, Japan). The inductively coupled plasma measurements were performed three times in each extraction.

**Cytotoxicity assay**. HEK-293EBNA cells (Invitrogen, current: Thermo Fisher, Waltham, MA, USA) were cultured in Dulbecco's modified Eagle's medium (Nacalai Tesque, Kyoto, Japan) containing 10% fetal bovine serum (Gibco, Grand Island, NY, USA), 250 μg/mL geneticin (Phyto Technology Laboratories, Shawnee Mission, KS, USA) and 100 μg/mL gentamicin (Life Technologies, Carlsbad, CA, USA). Prior to the cytotoxicity assay, the culture medium was replaced with Opti-MEM (Thermo Scientific). The cells were then treated with 1, 10, 100 μM, and 1 mM PDMA or DMA for 24 h at 37 °C. The cytotoxicities of PDMA and DMA were estimated by measuring the intracellular ATP content and the lactate dehydrogenase activity in the medium. The ATP content was measured using an ATPLite Kit (Perkin-Elmer Life and Analytical Sciences, Boston, MA, USA), in accordance with the manufacturer's instructions. Briefly, $2 \times 10^4$ cells/well in a 96-well plate were treated with Mammalian Cell Lysis Solution and incubated for 5 min at room temperature. Next, substrate buffer solution was added and the plate was incubated for 5 min at room temperature with shaking. ATP content was calculated as the percentage of luciferase chemiluminescence relative to that of mock-treated cells (control) using an Infinite M200 spectrophotometer (Tecan, Mannedorf, Switzerland). Lactate dehydrogenase activity was measured using a Cytotoxicity LDH Assay Kit-WST (Dojindo Laboratories, Kumamoto, Japan), in accordance with the manufacturer's instructions. Briefly, the supernatant of $2 \times 10^4$ cells/well was treated with working solution in a 96-well plate and incubated for 30 min at room temperature shielded from light. Subsequently, stop solution was added. Lactate dehydrogenase activity was calculated as the optical density at 490 nm relative to that of total cell lysate as 100%, and of the mock control as 0%, using an Infinite M200 spectrophotometer (Tecan). As positive controls, cells were treated with 300 μM or 1 mM H$_2$O$_2$ for 24 h at 37 °C. Assays were performed in triplicate and significant differences was evaluated by the Dunnett test.

**Reporting summary**. Further information on research design is available in the Nature Research Reporting Summary linked to this article.

## Data availability
The main data supporting the finding of this study are available within the paper and its Supplementary Information file. Source data are provided with this paper. Other relevant data are available from the corresponding author upon reasonable request. Source data are provided with this paper.

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

## Acknowledgements

We thank to Drs. P. Blamey and S. Mori for helpful discussions for plant biology. We are grateful to Drs. Elsbeth L. Walker for providing the ZmYS1 cDNA and Atsushi Yamagata for providing the insect cells expression vector and suggestions. We also thank to Drs. H. Imagawa and I. Okamoto (Tokushima Bunri University) for high-resolution mass spectra measurements of 6-membered analog of DMA. We would like to thank Drs. Hideo Saji and Hiroyuki Kimura from Kyoto University for discussions and use of radioisotopes at the Radioisotope Research Center. We also thank Ms. Kayoko Kousaka for excellent technical assistance, and Mrs. Mitsuo Hashimoto, Rintaro Yuki and Keisuke Maeda (Ishikawa Prefectural University) for assistance in field experiment. The support of biodegradation assay by Japan Food Research Laboratories is also acknowledged. This work was partially supported by JSPS KAKENHI Grant Numbers JP18H04416 in Middle Molecular Strategy,

JP16H03292, JP19H02851, JST value program Grant Number VP29117941288, Grants-in-Aid for Scientific Research (Grant No. 21310148), and financial support of the Research Clusters program of Tokushima University (No. 1703015).

## Author contributions

K.N. and M.S. conceived the experiments and analyzed the results. A.U., S.S., R.T., S.N., H. Mukaiyama., A.M., M.K., K.K., B.B.S., and A.N. performed the laboratory experiments and optimized the reaction conditions for the synthesis of mugineic acid analogs. Y.M. performed the in vitro assay of synthetic analogs. H.N. performed the qPCR experiment. H. Masuda., M.S.A., and T.K. performed pilot field experiments and metal concentration measurement. A.U. and M.T. determined the stability constant of PDMA for Fe(III). K.F. and H.F. performed the cytotoxicity assay of PDMA. Y.C. and H.N. performed the HRMS analysis of the xylem sap of rice plants. T.W. performed HRMS experiments of iron complexes with synthetic analogs. M.S. and K.N. wrote the paper, and T.K., K.T., and N.K.N. proofread the paper.

## Competing interests

M.S. and A.M. are employed by the company AICHI STEEL CORPORATION. The remaining authors declare no competing interests.
