## [Peer Review File · Nature Communications]

REVIEWER COMMENTS

Reviewer #1 (Remarks to the Author):

This is a paper on development of a Fe fertilizer. Given that other Fe fertilizers have been developed and applied widely, it is necessary to show the advantage of this fertilizer in terms of effect, cost, yield, etc. This requires to be tested in different fields using different crops for multiple years. However, the authors only tested in pot experiment using young seedlings. Furthermore, they used rice as a testing plants. As mentioned before, Fe-deficiency in rice is not a problem worldwide.

Overall, this work is still at premature stage.

Reviewer #2 (Remarks to the Author):

I am pleased to see that the authors have finally correlated SPAD values to iron concentrations in the leaves. This was a missing piece of evidence to show that PDMA really improves iron uptake, rather than simply improves plant growth for some other reason. This evidence is required to support the title “novel iron fertilizer” rather than “novel fertilizer”.

This is the minimal evidence that is required, and I see that the error bars on the Fe measurements in Fig 6 are quite high compared to those for other elements. Do the authors have an explanation for this? Bio-markers of iron status in the plant, such as ferritin protein levels using Western blots, or expression levels of certain genes, could provide further evidence for direct alleviation of Fe deficiency, but I will leave that to the authors if they have the capability to carry out these experiments.

Abstract:

Line 17: reduced

Line 25-26: tone down the end of the sentence, and certainly delete “prevent or”. See also comment regarding opening paragraph of the Discussion.

Introduction:

Line 41: transgenic lines overexpressing genes...

Results: further issues with English, perhaps the Editorial office can address this

Fig 2 legend and Results text is unclear. Were the rice plants already iron deficient when transplanted? Please state clearly that the SPAD value correlates to chlorosis, a major symptom of iron deficiency. Also state typical values for iron-deficient rice and iron-sufficient rice.

Line 191: what is meant by typical peak value?

Line 194 – 195: This demonstrated superior Fe uptake by rice treated with Fe–PDMA and Fe–DMA than with Fe–EDDHA: you cannot say this at this point, because SPAD values have not been linked to Fe uptake. SPAD only measures “green colour”.

Figs 5 and 8, labels: space between number and unit.

Fig 6, legend: it says 3 leaves of each pot, and later is says n = 10. Is it correct there were

10 pots per treatment? Please state this more clearly.

Discussion:

First paragraph is overstated; Fe deficiency in crops is a minor problem compared to drought, diseases etc. Also, most of the world's rice crop is grown in flooded paddies, where iron toxicity rather than deficiency is the problem. Moreover, there are no data for grain yield in this paper, only for vegetative growth.

Re: Revision of our Manuscript NCOMMS-20-23306A-Z

Dear reviewer 2:

We deeply appreciate kind suggestions and corrections of the reviewer 2 for making the revised manuscript more valuable. We have further revised the manuscript according to the suggestions of reviewer 2 as explained below. Also, we attached the 'track changes' version, which highlighted the revised and rewritten parts, for easy review. The yellow highlights indicate the revised parts according to the suggestion of the reviewer 2, and the blue highlights indicate the modified parts according to formatting instructions.

Our responses to the suggestion of Reviewer 2:

[About high error bars on the Fe measurements (Fig. 6)]:

The reason for high error bars in the Fe measurement is due to outliers. Although some outliers were observed in applications of PDMA, Fe-PDMA, Zn-PDMA, and Fe-EDTA, we showed the data as is. To easily recognize the existence of outliers and clearly show the distribution of the data, we replaced bar graphs with bar graphs with plots in this revision according to the suggestion of the editor.

About outliers: These outliers (< 10 ppm) should indicate a tremendous iron deficiency, but rice did not show such symptoms. Thus, it is possible that only Fe was removed for some reason such as aggregation and precipitation of Fe during the pretreatment for ICP measurements.

In addition, we appreciate the advice of reviewer 2 that the check of bio-markers of iron status will support our experimental results. As shown in Supplementary Fig. 7, we have confirmed the expression of genes *OsNAS1* and *OsIRO2* that are used as biomarkers of iron deficiency. The application of PDMA suppressed the expression of *OsNAS1* and *OsIRO2*, while the applications of FeCl₃ and EDTA-Fe complex only slightly reduced them, demonstrating that PDMA significantly recovers iron deficiency.

We have confirmed the expression of Fe deficiency biomarkers as described above, but if we do not respond correctly to the meanings suggested by reviewer 2, please let us know.

[Other points]:

Abstract:

1. *Line 17: reduced*

=> “reducing” was corrected to “reduced”.

2. *Line 25-26: tone down the end of the sentence, and certainly delete “prevent or”. See also comment regarding opening paragraph of the Discussion.*

=> We have toned down the end of the sentence and briefly summarized the contents of manuscript. Also, we removed the description of the food crisis issue throughout the manuscript.

Introduction:

3. *Line 41: transgenic lines overexpressing genes...*

=> We thank the Reviewer for this suggestion. With consideration, we have chosen the expression “harboring highly efficient genes”, because of the following reason. Some of the transgenic rice tolerant to Fe deficiency were introduced with barley genomic fragments, including promoter regions and corresponding genes participated in MAs synthesis, which are not necessarily overexpressed in whole plant, but rather regulated spatially and induced by Fe deficiency in roots. In addition, the activities of the enzymes encoded by barley genes are sometimes higher than corresponding rice genes. Therefore, we use the expression “harboring highly efficient genes”. Of course, if the reviewer or the Editor consider that “overexpressing” is better description, we will follow that consideration.

Results

4. *further issues with English, perhaps the Editorial office can address this*

=> Thank you for your appropriate advice. Although our manuscript has been proofread by an English editing company at the first submission, we greatly appreciate if the Editorial office addresses this issue. Of course, we will ask the editing company again if necessary.

5. *Fig 2 legend and Results text is unclear. Were the rice plants already iron deficient when transplanted? Please state clearly that the SPAD value correlates to chlorosis, a major symptom of iron deficiency. Also state typical values for iron-deficient rice and iron-sufficient rice.*

=> 0 day in Fig. 2a means the day of final application of chelates (14 days after transplanting). We have described the experimental details more clearly in the main text. In addition, we have stated

clearly that the SPAD value correlates to chlorosis, and also stated typical values for iron-deficient rice and iron-sufficient rice.

6. *Line 191: what is meant by typical peak value?*

=> We have added that the typical value means the darkest green rice leaves.

7. *Line 194-195: This demonstrated superior Fe uptake by rice treated with Fe-PDMA and Fe-DMA than with Fe-EDDHA: you cannot say this at this point, because SPAD values have not been linked to Fe uptake. SPAD only measures "green colour".*

=> Following the suggestion of the reviewer 2, we have removed the mention of Fe uptake and instead described it as a growth effect.

8. *Fig. 5 and 8: labels: space between number and unit.*

=> Space between number and unit was added.

9. *Fig. 6, legend: it says 3 leaves of each pot, and later it says $n = 10$. Is it correct there were 10 pots per treatment? Please state this more clearly.*

=> Yes, there were 10 pots per treatment. We have revised the corresponding description to state more clearly about samples of ICP-OES measurements.

Discussion:

10. *First paragraph is overstated; Fe deficiency in crops is a minor problem compared to drought, diseases etc. Also, most of the world's rice crop is grown in flooded paddies, where iron toxicity rather than deficiency is the problem. Moreover, there are no data for grain yield in this paper, only for vegetative growth.*

=> Following the suggestion of the reviewer 2, we have removed most of the first paragraph to avoid overstatements.

We thank again the reviewer 2 for careful proofreading of our manuscript, and we hope that this revision reaches the level required by the reviewer 2.

Sincerely yours.

Kosuke Namba

REVIEWERS' COMMENTS

Reviewer #2 (Remarks to the Author):

I am pleased to see the willingness of the authors to further improve their manuscript "Development of a mugineic acid family phytosiderophore analog as an iron fertilizer".

For the error bars in Fig 6b, I welcome seeing the individual data points in the graph. Based on this, the outliers for Fe can be removed, as there are only 1 or 2 outliers, versus 8 - 9 data points that are close together.

For Fig 6, note the letters b - e are missing in the figure and should be added.

I am also pleased to see gene expression on OsNAS2 and OsIRO2, used as marker genes for Fe deficiency in rice. Please verify the primers were against OsNAS2, because in your Response Letter you say OsNAS1.

The description of the experimental set up of rice growth experiments in Fig 2 and elsewhere is now made sufficiently clear, at least for me.

Regarding the English, there are many small issues, both in semantics and grammar. Mostly, this does not affect the meaning, but I often had to read sentences or small sections twice. I suggest the editing service used by the authors is not used again, but they find another company.

For line 46: the adjective 'efficient' cannot be used for genes, one needs to specify what the genes are efficient for. In this case, it would be iron uptake. The sentence continues with "involved in..." to explain more about these genes. Having added that, one can simply say "harboring genes involved in ..."

Re: Revision of our Manuscript NCOMMS-20-23306B

Dear reviewer 2:

We deeply appreciate kind suggestions and corrections of the reviewer 2 for making the revised manuscript more valuable. We have further revised the manuscript according to the suggestions of reviewer 2 as explained below. Also, we attached the 'track changes' version, which highlighted the revised and rewritten parts, for easy review. The yellow highlights indicate the revised parts according to the editorial request, and the blue highlights indicate the modified parts according to English editing service.

Our responses to the suggestion of Reviewer 2:

For the error bars in Fig 6b, I welcome seeing the individual data points in the graph. Based on this, the outliers for Fe can be removed, as there are only 1 or 2 outliers, versus 8 - 9 data points that are close together.

==> Thank you for the kind suggestion. According to your suggestion, 1 or 2 outliers in Fig 6b were removed to demonstrate more accurate data. The SPAD values and metal concentrations corresponding to outliers in Fig 6b were also removed from Fig 6a, c, d, and e. Then, the graphs in Fig 6 were remade without outliers, and providing p-values in the graph and legend according to Editorial requests. As a result, slight differences in statistical significance occurred (such as Fe-EDTA treatment vs. control in Fe concentration, or PDMA treatment vs. control in Cu concentration), but this revision has not influenced the main results and story of this manuscript.

For Fig 6, note the letters b - e are missing in the figure and should be added.

==> Thank you for the careful proofreading, labels b-e were added to Fig 6.

I am also pleased to see gene expression on OsNAS2 and OsIRO2, used as marker genes for Fe deficiency in rice. Please verify the primers were against OsNAS2, because in your Response Letter you say OsNAS1.

==> Thank you for the careful proofreading. As shown in primers list (supplementary table 3), we actually adopted OsNAS2.

The description of the experimental set up of rice growth experiments in Fig 2 and elsewhere is now made sufficiently clear, at least for me.

==> Thank you for the acceptance.

Regarding the English, there are many small issues, both in semantics and grammar. Mostly, this does not affect the meaning, but I often had to read sentences or small sections twice. I suggest the editing service used by the authors is not used again, but they find another company.

==> We asked another editing service again. The rewritten parts according to the editing service are shown in blue highlight.

For line 46: the adjective 'efficient' cannot be used for genes, one needs to specify what the genes are efficient for. In this case, it would be iron uptake. The sentence continues with "involved in..." to explain more about these genes. Having added that, one can simply say "harboring genes involved in ..."

==> Thank you for the adequate advise, we removed 'highly efficient' and the sentence 'the synthesis of MAs' was continued with "involved in...".

We thank again the reviewer 2 for careful proofreading of our manuscript.

Sincerely yours.

Kosuke Namba